# Pine Pitch Canker and Insects: Relationships and Implications for Disease Spread in Europe

**Mercedes Fernández-Fernández** [1,2,*], **Pedro Naves** [3], **Johanna Witzell** [4],
**Dmitry L. Musolin** [5], **Andrey V. Selikhovkin** [5,6], **Marius Paraschiv** [7], **Danut Chira** [7],
**Pablo Martínez-Álvarez** [2,8], **Jorge Martín-García** [8,9], **E. Jordán Muñoz-Adalia** [10],
**Aliye Altunisik** [11], **Giuseppe E. Massimino Cocuzza** [12], **Silvia Di Silvestro** [13],
**Cristina Zamora** [2,8] and **Julio J. Diez** [2,8]

1   Department of Agroforestry, ETSIIA Palencia, University of Valladolid, Avenida de Madrid 44,
    34071 Palencia, Spain
2   Sustainable Forest Management Research Institute, University of Valladolid-INIA, Avenida de Madrid 44,
    34071 Palencia, Spain
3   Instituto Nacional de Investigação Agrária e Veterinária, INIAV, Av. da República, Quinta do Marquês,
    2780-505 Oeiras, Portugal
4   Swedish University of Agricultural Sciences. Southern Swedish Forest Research Centre,
    P.O. Box 49 SE-230 53 Alnarp, Sweden
5   Department of Forest Protection, Wood Science and Game Management, Saint Petersburg State Forest
    Technical University, Institutskiy per., 5 St. Petersburg 194021, Russia
6   Department of Biogeography and Environmental Protection, Saint Petersburg State University.
    13B Universitetskaya Emb., St. Petersburg 199034, Russia
7   National Institute for Research and Development in Forestry "Marin Dracea", Brașov Research Station,
    13 Closca Str., 500040 Brașov, Romania
8   Department of Plant Production and Forest Resources, University of Valladolid, Avenida de Madrid 44,
    34071 Palencia, Spain
9   Department of Biology, CESAM (Centre for Environmental and Marine Studies), University of Aveiro,
    Campus Universitario de Santiago, 3810-193 Aveiro, Portugal
10  Forest Sciences Center of Catalonia (CTFC), Carretera St. Llorenç de Morunys, 25280 Solsona, Lleida, Spain
11  Department of Plant Protection-Phytopathology, Akdeniz University, 07070 Campus Antalya, Turkey
12  Department of Agriculture, Food and Environment, University of Catania, Via Santa Sofia, 100, 95123
    Catania, Italy
13  Council for Agricultural Research and Economics (CREA), Research Centre for Olive, Citrus and Tree Fruit,
    Corso Savoia 190, 95024 Acireale, Italy
*   Correspondence: mffernan@agro.uva.es

**Abstract:** The fungal pathogen *Fusarium circinatum* (Nirenberg and O' Donnell) is the causal agent of pine pitch canker (PPC) disease, which seriously affects conifer species in forests and nurseries worldwide. In Europe, PPC is only established in the Iberian Peninsula; however, it is presumed that its range could expand through the continent in the near future. Infection caused by this fungus requires open wounds on the tree, including physical damage caused by insects. Therefore, a relationship probably occurs between PPC and a wide variety of insects. The aim of this review is to outline the taxonomic and ecological diversity of insect species with high potential association with *F. circinatum* in Europe and elsewhere. The insects were classified as vectors, carriers and wounding agents according to the association level with the PPC disease. In addition, we discuss the insect-mediated spreading of PPC disease in relation to the different phases of forest stand development, from seeds and seedlings in nurseries to mature stands. Lastly, to improve our predictive capacities and to design appropriate intervention measures and strategies for controlling disease dissemination by insects, variables such as geographic location, time of the year and host species should be considered.

Our review provides a framework of the multiple factors that regulate the insect–host interactions and determine the success of the infection.

**Keywords:** pine pitch canker; vectors; carriers; wounding agents; forests; nurseries; insect control

## 1. Introduction

The ascomycete fungus *Fusarium circinatum* (Nirenberg and O' Donnell) is the causal agent of pine pitch canker (PPC), a serious disease that has been recorded in more than 50 species of pines (*Pinus* spp.) and *Pseudotsuga menziesii* (Mirb.) Franco [1]. Native to North America, this pathogen has been found in South Africa, Japan, Korea, Mexico, Chile, Uruguay, Colombia, Brazil as well as in Europe (France, Spain, Italy, and Portugal [1]). The fungus is currently included in the A2 list of quarantine organisms (recommended for regulation) of the European and Mediterranean Plant Protection Organization (EPPO) and has so far limited distribution in the European and Mediterranean regions [1,2].

*Fusarium circinatum* is a pathogen that causes high seed mortality [2]. It can also cause necrosis, chlorosis, wilting of needles, dieback, desiccation, and eventual death of seedlings [3]. In mature trees, it causes pitch resin-soaked cankers in trunks and large branches, which can girdle and predispose the trees to being broken during windstorms. The extent of damage varies among regions, host species, and habitats ranging from plantations, nurseries, parks and gardens or forests [4]. The pathogen can be found in all parts of the host trees (root, trunk, branches, shoots, cones, and seeds), as well as in the forest litter. The asexual spores (microconidia and macroconidia) are produced in a viscous liquid, but can also become airborne.

As a wound pathogen, *F. circinatum* needs mechanical wounds or feeding/entrance holes of insects to enter the plant, and its spores may be attached to motile organisms. It is thus not surprising that studies conducted in different countries have demonstrated that forest insects associated with pines play an important role in the natural spreading of PPC (e.g., [1,4]). Some insect species act as vectors (carrying and transmitting the pathogen to the trees) [4–6], carriers (only carrying the pathogen) and others, as wounding agents (providing infection court for the pathogen). The assessment of the insects associated with *F. circinatum* is a dynamic exercise, as the nature of the association can vary between geographic locations, time of the year and pine hosts. Multiple factors regulate the association between the insect and the pathogen: for instance, the ecology and life cycle of the insects and their association with the host tree are crucial, as insects feeding on fresh tissues and requiring frequent visits to living pines are more prone to successfully inoculate the fungal spores into healthy hosts. The insect´s mobility is also important, and adults of the majority of the species associated with *F. circinatum* can actively move and disperse it by flight, resulting in a higher risk of disseminating the pathogen within and between forests. Finally, another fundamental aspect is the ability of the insects to transport the spores of the fungi on their body, which conditions the phoretic rate, i.e., the proportion of insects carrying the pathogen, and which is dependent of a complex interaction between environmental factors and anatomical characters of the insect.

Given that *F. circinatum* was only recently recorded for the first time in Europe [1], it is foreseeable that its range can noticeably expand through the continent in the near future. To control or locally eradicate the pathogen, it is critically important to have detailed knowledge on which European insects can promote spreading of the fungus and how they can do it. In this review, we analyze the taxonomic and ecological diversity of insect species with known or high potential association with *F. circinatum* in Europe and elsewhere, and critically asses their potential risk as vectors *sensu lato*.

## 2. Interactions of *Fusarium circinatum* with Insects

Worldwide, *F. circinatum* has been reported to be phoretically associated with a broad array of insect species [1,4]. Some of these are thought to be potential vectors of *F. circinatum* (meaning that in addition to carrying the pathogen, and visiting susceptible plant hosts, such species are capable of successfully transmitting the pathogen to trees that were not yet infected). Others are assumed to be carriers (which means that they carry the pathogen and visit susceptible hosts, but their capacity to transmit the pathogen has not yet been demonstrated). A third group of insects, which can contribute to PPC epidemiology, are the wounding agents, which create infection courts for the pathogen, by damaging the bark and wood with their tunnels and entrance holes, and/or feeding on shoots, twigs and cones [4]. Moreover, the wound has to be deep enough to provide adequate moisture; the spores need to germinate before wound desiccation, provided that not too much resin has been produced to stop the process [7,8].

Altogether, these groups of insects are taxonomically and ecologically heterogeneous and include, among others, shoot and foliar feeders (Coleoptera: Curculionidae; Lepidoptera: Tortricidae), bark beetles and wood borers (Coleoptera: Curculionidae), root borers (Coleoptera: Curculionidae), cone insects (Lepidoptera: Tortricidae, Pyralidae; Hemiptera: Coreidae, Scutelleridae; Coleoptera: Curculionidae, Anobiidae), sucking insects (Hemiptera: Aphrophoridae, Cercopidae) and predators (Coleoptera: Zopheridae, Cleridae; Diptera: Dolichopodidae).

The assessment of the insects associated with *F. circinatum* as vectors, carriers or wounding agents is a challenging exercise, as the nature of the association can vary between geographic location, time of the year and host species. Multiple factors regulate the association between the insect and the pathogen: for instance, the spatiotemporal synchrony of ecological traits and life cycle of the insects and host trees are crucial [9]. The feeding habit is also important, and insects feeding on fresh tissues and requiring frequent visits to living pines are more prone to successfully inoculate the fungal spores into healthy hosts. Another important trait is the insect´s mobility, and the more active the species associated with *F. circinatum*, the higher the risk of disseminating the pathogen within and between forest stands [4]. Finally, a fundamental aspect is the ability of the insect to transport the spores of the fungi on their body, which conditions the phoretic rate, i.e., the proportion of insects carrying the pathogen, and being dependent on a complex interaction between environmental factors and anatomical characters of the insect [1,2,4].

### 2.1. Interactions of F. circinatum with Insects in European Countries

*Fusarium circinatum* has been reported to be phoretically associated to various species of bark beetles (Coleoptera: Curculionidae) in *Pinus radiata* (D. Don) plantations in northern Spain, namely *Pityophthorus pubescens* (Marsham, 1802), *Hylurgops palliatus* (Gyllenhal, 1813), *Ips sexdentatus* (Börner, 1776), *Hypothenemus eruditus* Westwood, 1836, *Hylastes attenuatus* Erichson, 1836, *Orthotomicus erosus* (Wollaston, 1857), *Pissodes validirostris* (C.R. Sahlberg, 1834), *Brachyderes incanus* (Linnaeus, 1758) [10], and *Tomicus piniperda* (Linnaeus, 1758) [11]. In general, the biology of most of these species is well studied, as some are important forest or nursery pests (e.g., [12–17]). The nature of their association with *F. circinatum* as well their preferred hosts and distribution range are summarized in Table 1.

**Table 1.** Ecological characteristics and distribution of insects associated with *Fusarium circinatum* in Europe considered as vectors, carriers or wounding agents.

| Species | Association with *F. circinatum* and Possible Transmission Pathways | Preferred Pine Habitat | Distribution Range | Selected References |
|---|---|---|---|---|
| **Shoot and foliage feeders** | | | | |
| *Tomicus piniperda* | Vector; shoots (feeding); bark and phloem (oviposition) | Pine stands of different ages | Europe, North-Western Africa and Northern Asia; introduced in North-eastern North America (USA and Canada) | [11,18] |
| *Rhyacionia* spp. | Wounding agent; shoots (oviposition) | Young pine stands | Widespread in Europe Asia, Northern Africa, North America; introduced in South America | [12,19] |
| **Bark beetles and wood borers** | | | | |
| *Pityophthoruspubescens* | Carrier; bark and phloem of mainly small branches (twigs) (oviposition and feeding) | Pine stands of different ages | Europe, Asia minor, Northern Africa (Morocco) | [10,20–23] |
| *Hylurgops palliatus* | Carrier; bark and phloem of trunk, branches and occasionally roots (oviposition and feeding) | Pine stands of different ages | Europe, Northern Asia, Northern Africa; introduced in North America | [10,24,25] |
| *Ips sexdentatus* | Carrier; bark and phloem of trunks (oviposition and feeding) | Mature pine stands | Europe, Northern Asia and East to China | [10,26,27] |
| *Hypothenemus eruditus* | Carrier; bark and phloem of small twigs, even in leaf petioles (oviposition and feeding), occasionally in seeds and fruits | Pine stands of different ages | Widespread in tropical, subtropical and temperate regions of the world; introduced in some European countries | [10,28,29] |
| *Hylastes attenuatus* | Carrier; bark and phloem of branches, trunk, stumps and roots | Pine stands of different ages | Europe, Northern Africa, Asia and East to China, Korea and Japan | [4,10,25,30,31] |
| *Orthotomicus erosus* | Carrier; bark and phloem of trunk and branches (oviposition and feeding) | Pine stands of different ages | Europe, Asia, Northern Africa; introduced in North and South America, Southern Africa, and Oceania | [1,10,24,30] |
| **Root borers** | | | | |
| *Brachyderes incanus* | Carrier; bark and phloem of branches, roots | Young pine stands | Europe, introduced in North America (USA and Canada) | [10,32] |
| **Cone insects** | | | | |
| *Pissodes validirostris* | Wounding agent, carrier; cones and shoots (feeding of beetles); cones (oviposition) | Mature pine stands | Europe, Northern Asia and East to China | [33–35] |

### 2.1.1. Shoot and Foliage Feeders

The genus *Tomicus* (Coleoptera: Curculionidae) has eight species worldwide, with three occurring in Europe: *Tomicus minor* (Hartig, 1834), *T. piniperda* and *T. destruens* (Wollaston, 1865) and are amongst the most important pests of pine forests. In Spain, *T. piniperda* is the most plausible vector of PPC [11] and, together with *T. destruens*, these species are widely distributed in Iberia Peninsula and Portugal [18,36].

The main hosts of European *Tomicus* are pines (*Pinus* spp.), including non-native species such as *P. radiata* and *Pinus strobus* Linnaeus [18,37]. Although these insects can attack apparently healthy trees, they prefer stressed pines, especially scorched and burnt trees [38,39].

*Tomicus* species occupy different niches in their hosts during their life cycle; while the preadult stages develop subcortically in the phloem, the adults perform maturation and regeneration feedings within the shoots of healthy pines (hence the common name of "pine shoot beetles"). The hibernation period continues within the shoots of the pine crown in the Southern latitudes, in the base of the pine trunks or in the soil litter in northern latitudes [21,40]. For the maturation and regeneration feeding, the adults fly to the crown of healthy pines and excavate galleries into the pith of the current-year shoots, which eventually break after windy or rainy conditions and fall to the ground. Shoot feeding can cause big damages when attacks occur in young plantations or when a large proportion of shoots is destroyed, resulting in loss of growth and pine decline, making the trees suitable for trunk attack by the following beetle generation or by secondary beetles [18].

Several fungal species have been described as associated with *Tomicus* beetles, including species of *Ophiostoma* and *Leptographium* [18,41,42]. Despite these consistent associations, *Tomicus* beetles apparently do not have specialized structures, such as mycangia, to transport the fungi [43], although structures present in the base of setae could act as fungi transport frames [1]. Adults can also carry nematodes, including species of the *Bursaphelenchus* genus [18].

In Europe, it was concluded, according to Leach´s postulates, that *T. piniperda* is a plausible vector of PPC, although with a low phoresy rate [11]. The low phoresy rate could be explained because in

*P. radiata* terpene expression triggered by *T. piniperda* inhibits *F. circinatum* growth [44]. In other studies, propagules of *F. circinatum* were not obtained from insects collected from infected stands [10,45,46]. Nevertheless, *Tomicus* beetles should be considered important vectors of PPC in Europe due to their widespread distribution and their status as pests in pine forests. In addition, their particular behavior consisting in feeding on several shoots/insect creates feeding wounds in healthy tree that determines their potential importance in the disease spreading. Moreover, they are effective vectors/carriers of fungi and other organisms.

Pine tip moths belonging to the genus *Rhyacionia* (Lepidoptera: Tortricidae) are also shoot feeders, and *Rhyacionia buoliana* (Denis and Schiffmüller, 1775) has been reported on more than 30 species of pines, including *P. radiata* [12]. Damage is caused by the larvae feeding in pine shoots and buds, with mining galleries usually accompanied by webbing and flow of resin [47]. Infested shoots and buds wilt and die, leading to deformation of the tree axis and affecting the tree shape and growth, which is particularly harmful in young plantations [48]. While this moth has not yet been found associated with PPC, they could be possible wounding agents, as it was mentioned for other species of the genus in USA.

There are other moths for which association with PPC has not yet been demonstrated. However, due to their widespread distribution, their pest status and their ability to create feeding wounds in healthy trees, they could be possible carriers and, most likely, wounding agents.

Another insect to consider is the pine sawyer beetle *Monochamus galloprovincialis* (Oliver) (Coleoptera: Cerambycidae) that is a shoot-feeder as an adult, being generally considered a secondary wood-boring insect associated with weakened, scorched or recently dead pines for breeding [20,49]. However, it can also occasionally damage conifer forests following population outbreaks [50–52]. Furthermore, *M. galloprovincialis* is the vector of the introduced pinewood nematode, which has been causing significant pine mortality in both Portugal and Spain [20]. Several fungal species have been found associated with *M. galloprovincialis* [53], including pathogenic and blue-stain fungi such as *Ophiostoma minus* (Hedgc.) (Syd. and P. Syd.), *Ophiostoma piceae* (Münch) (Syd. and P. Syd.), *Leptographium procerum* (W.B. Kendr.) (M.J. Wingf.) and *Fusarium oxysporum* Schltdl. found on insects and galleries with a relative abundance below 1% [53,54]. While these beetles have not yet been found associated with PPC, they are possible vectors and, most likely, wounding agents, that would contribute to its dissemination.

### 2.1.2. Bark Beetles and Wood Borers

Bark beetles (Coleoptera: Curculionidae) associated with pines can generally be considered carriers of *F. circinatum*. Twig beetles (*Pityophthorus* spp.) (Coleoptera: Curculionidae) have been associated with PPC in Europe and North America. *Pityophthorus pubescens* is a non-aggressive forest insect, colonizing weakened trees or broken branches in healthy pines [22], where it constructs galleries in the phloem or in the pith of small branches [55]. In Spain, Romón et al. [10] found *F. circinatum* in 25% of the sampled *P. pubescens* adults, while Bezos et al. [22] reported a weaker association with lower phoresy rates (maximum of 2%) and low incidence of the pathogen in the insects' galleries. These variations may result from differences in the ambient humidity, which can condition the efficiency of *Pityophthorus* species in carrying the PPC pathogen [55], or from different population levels in the pine stands. In addition to *P. pubescens*, other European *Pityophthorus* species share similar morphological and ecological traits and live on conifers (mainly on pines, but also on *Picea* and *Abies* species), and therefore also may become associated with the PPC pathogen.

The genus *Ips* includes some of the most important forest pests of pines and spruces, with seven species occurring in Europe [26]. In Southern Europe, the most important species is *I. sexdentatus*, causing significant mortality in pine stands (*Pinus sylvestris* Linnaeus, *P. pinaster* and *P. radiata*) affected by fires, storms or droughts [30,56,57]. In Northern Spain, it is considered one of the most important pests affecting *P. radiata* in mature stands, where weakened trees are at particular risk [20,58]. The species is also shown to be a carrier of *F. circinatum* [10,59]. *Ips* beetles do not make wounds or galleries on

twigs or shoots of healthy pines, so the risk of being effective vectors of PPC is lower than for other similar bark beetles such as *Tomicus* species. However, *Ips* beetles can attack healthy pines (including *P. radiata*), inoculate fungi, fly for long distances and produce several generations per year in Southern Europe [60], suggesting high risk of becoming associated with *F. circinatum* in European pine forests. There is some possibility that the pine engraver bark beetle *I. acuminatus* might become a vector or carrier of *F. circinatum* in Europe. The species is considered a serious pest of pines and it is attracting growing attention in Ukraine, Belorussia, Russia, Finland and in some other countries [16,61,62]. It has been demonstrated that *I. acuminatus* can vector dozens of fungi including pathogens such as *Diplodia pinea* and *Ophiostoma minus* [63]. However, the role of this species in spreading PPC will likely be limited as this species has maturation feeding under the bark of exactly the same (and already weakened) trees on which they had just developed as larvae. *Ips acuminatus* can attack branches and tops of healthy trees, but it prefers to inhabit irreversibly weakened pines.

*Orthotomicus erosus* has also been found carrying spores of *F. circinatum* in Northern Spain [10]. This widespread bark beetle develops mainly on pines, including *P. radiata* [1,10,30], although it can also be found on other conifers [24,64–66]. Similarly to other bark beetles, *Orthotomicus* spp. are frequently found carrying fungi, particularly ophiostomatoid fungi [67–69], including species of *Fusarium* genus [10,25].

*Hylurgops palliatus* is another bark beetle also found carrying *F. circinatum* in Northern Spain [10]. It is a widespread forest species found throughout the Palearctic region [24,70], being generally considered a secondary bark beetle on stumps or freshly cut logs [71]. It is associated with several fungi species, particularly blue-stain fungi of the genera *Leptographium*, *Graphium* [25,72–74], and *Ophiostoma* [75].

*Hypothenemus eruditus* has been found in Northern Spain carrying propagules of *F. circinatum* [10]. It is present in all tropical and subtropical regions of the world, and in many temperate regions [29], including Spain, France, Italy, and Malta [76]. The adults are extremely small (only up to 1.3 mm), and are exceptionally polyphagous, having been recorded from hundreds of plant hosts [28,77]. The widespread distribution, morphological variations, extended habits, and host range suggest that *H. eruditus* may, in fact, represent a species-complex still to be discriminated [29].

Pine weevils of the genus *Pissodes* (Coleoptera: Curculionidae) are important forest pests, and *Pissodes castaneus* can cause significant damage to pines (and occasionally to larches and spruces) as larvae and adults [58,78–80]. Larvae develop on the inner bark of branches, trunks and even roots, making a characteristic pupal cell under the bark before emergence [78]. These species show frequent mutualistic relationships with blue-stain fungi [25], including pathogenic species of the genera *Leptographium* [81], *Armillaria* [82], and *Sporothrix* [83,84]. *Pissodes castaneus* have been found to be carriers of *F. circinatum* in Spain [10,30].

As with other similar bark beetles, species of the genus *Hylastes* are associated with several fungi species [25], particularly blue-stain fungi of the genus *Ophiostoma* (anamorph *Graphium*) [72] and *Leptographium* [85], which they transmit to pine seedlings by feeding [86]. *Hylastes* spp. are associated with PPC and have been found to be carriers of *F. circinatum* in North America [4] and in Spain, e.g., *H. attenuatus* [10]. In Europe, eleven species occur, mainly associated with pines [37,87], including *P. radiata* [30,86].

*Hylurgus ligniperda* (Fabricius, 1787), native to Central and Southern Europe, Asia Minor and Algeria [87]. Its hosts are species of the family Pinaceae. It breeds on the bark of dead and dying trees, usually in the thick sections near the base of the trunk, stumps and in large exposed roots [88]. It is considered a secondary pest, not causing significant damage to pine forests [37]. Nevertheless, it is known to transmit blue-stain fungi to wood via its galleries, along with root pathogens [89–92]. *Hylurgus* beetles have not been found to be associated with PPC in Europe or North America [4,10], but should be considered possible carriers keeping in mind their ecology, life habits and frequent association with decaying pine hosts and fungi.

*Hylobius abietis* (Linnaeus, 1758) (Coleoptera: Curculionidae) is an important pest in Scandinavia, Northern European part of Russia and Eastern Siberia, affecting particularly one- to three-year-old coniferous plantations and regeneration areas [93,94]. It can be found on *P. radiata* [25], and is considered a pest of significant concern for plantations in Northern Spain [58]. Moreover, it is an important vector of numerous fungi (e.g., [84]), and although it has not been found associated with *F. circinatum*, a plausible association as carrier cannot be discarded.

### 2.1.3. Root Borers

*Brachyderes incanus* (Coleoptera: Curculionidae), is considered a minor conifer pest, which is important only in newly established plantations [32], although it has been found carrying *F. circinatum* in Northern Spain [10]. Larvae bore into and strip bark from roots of the hosts, occasionally killing roots up to 3 mm in diameter, while the adults feed on conifer needles and occasionally damage the bark [32].

### 2.1.4. Cone Insects

European insects associated with cones can be considered carriers or wounding agents that facilitate *F. circinatum* infections. *Pissodes validirostris* (Coleoptera: Curculionidae), is the only member of its genus feeding on the seed cones of pines [95], and in the Iberian Peninsula it is an important pest of *Pinus pinea* Linnaeus cones, causing economic losses [33]. It was considered a biological control agent against invasive pines in South Africa, and the relationship between *P. validirostris* and *F. circinatum* was investigated under controlled conditions [35]. The authors found that, under controlled conditions, the weevils were not able to transmit the fungus, but the feeding wounds favored the infection of the host by *F. circinatum*.

Another cone insect is *Leptoglossus occidentalis* (Heidemann, 1910) (Hemiptera: Coreidae), native to North America but accidentally introduced into Europe [96]. Adults feed on the young seeds or flowers of over 40 species of conifers, mainly pines, causing infertility or total destruction of the seeds [97–99]. In the Mediterranean Basin, this seed bug has been related to the strong reduction of pine seed production, with a negative impact on the production of edible pine-seeds in *P. pinea* [100–103]. Despite the economic impact of this species, little is known about its associated fungal community, but Luchi et al. [104] reported a phoretic relationship with the pine pathogen *Diplodia pinea* (Desm.) in Mediterranean pine forests. Its widespread distribution, abundance and polyphagy makes *L. occidentalis* a plausible wounding agent of PPC in Europe, similarly to *Leptoglossus corculus* (Say, 1832) in North America [4].

Another important cone insect due to the high losses of pine-seeds of *Pinus pinea* in Spain is *Dyorictria mendacella* (Staudinger, 1859) (Lepidoptera, Pyralidae) [105]. Although no evidence of any relationship between PPC disease and this moth currently exist, it could be also considered as an important wounding agent and an entry point for the disease.

### 2.1.5. Sucking Insects

Even though there are no direct evidence of relationship between *F. circinatum* and sucking insects in Europe, some of the latter can potentially be at least wounding agents or carriers as they do make holes while feeding and can fly or can be carried by wind from one tree to another. The following species of sucking insects could deserve some attention: scale insects such as *Leucaspis* spp. (Hemiptera: Diaspididae) (in particular, *Leucaspis pusilla* (Loew, 1883), *Leucaspis lowi* (Colvée, 1882) and *Leucaspis pini* (Hartig, 1839)) [106], aphids of the genus *Cinara* (Hemiptera: Aphididae), in particular, *Cinara acutirostris* (Hille Ris Lambers, 1956), *Cinara brauni* (Börner, 1940), *Cinara pinea* (Mordvilko, 1894), *Cinara pinihabitans* (Mordvilko, 1896) and *Cinara schimitscheki* (Börner, 1940). Moreover, bark bugs, such as *Aradus cinnamomeus* Panzer, 1806 (Hemiptera: Aradidae) or spittlebugs like *Haematoloma dorsatum* (Ahrens, 1812) (Hemiptera: Cercopidae) could also have some participation in the PPC spreading in Europe [107–112] although these associations should be further clarified.

2.1.6. Predators

The possible association of *F. circinatum* with predatory insects has not been assessed in Europe. Yet, these insects could be possible carriers of the PPC pathogen, given their habits of predating on bark beetles and other insects under the bark and in galleries, where they may get in contact with *F. circinatum* propagules, although their overall importance as dispersal agents is expected to be low.

*2.2. Interaction of Fusarium circinatum with Insects in Non-European Countries*

As noted above, *F. circinatum* has been found in USA, South Africa, Japan, South Korea, Mexico, Chile, Uruguay, Colombia, and Brazil. However, to our knowledge, the available information regarding insects interacting with the pathogen outside of Europe is limited to studies carried out in USA, and, to a lesser extent, in Chile and South Africa. On the other hand, although *F. circinatum* has not been reported in Oceania [4,113], some potential vectors of PPC were reported in New Zealand. Taking into account that these authors already collated this information and no new relevant data are available, this review will focus only on the *F. circinatum*–insect interactions in those countries where PPC is already presented.

2.2.1. Shoot and Foliage Feeders

In North America, the shoot-feeders are mainly considered wounding agents, including the pine tip moths of the genus *Rhyacionia* (Lepidoptera: Tortricidae). Blakeslee et al. [114] and Dwinell et al. [115] pointed out that airborne spores of *F. circinatum* can readily infect wounds created by the subtropical pine tip moth (*Rhyacionia subtropica* Miller, 1961) and other similar species. Other authors isolated the PPC pathogen from shoots damaged by pine tip moths, and from larvae and pupae [116,117]. Runion et al. [118] reported that in eastern North Carolina, the percentage of *Pinus taeda* Linnaeus seedling terminals damaged by pine tip moths was positively correlated with infection by the PPC pathogen.

Other shoot feeders include *Contarinia* spp. flies (Diptera: Cecidomyiidae), which cause wounds on pines (particularly on *P. taeda* and *Pinus elliottii* (Engelm.)) in seed orchards and plantations in the SE of the USA [115]. According to Dwinell et al. [115], the wounds are often colonized by the PPC pathogen.

2.2.2. Bark Beetles and Wood Borers

Species of the genus *Pityophthorus* (Coleoptera: Curculionidae) are the most studied insects as agents of dispersion of the PPC disease outside Europe. Most of the studies have been performed in California (USA), where at least four species (*Pityophthorus setosus* (Blackman, 1928); *Pityophthorus carmeli* (Swaine, 1918); *Pityophthorus nitidulus* (Mannerheim, 1843); and *Pityophthorus pulchellus* (Eichhoff, 1869)) have been shown to act as carriers of the pathogen *F. circinatum* [119,120]. Storer et al. [6] demonstrated that *P. setosus* acts as a vector of *F. circinatum* in an experiment in which the insects were artificially contaminated with spores of the pathogen and caged with branches of *P. radiata.* It was found that trees baited with insect pheromones were more likely to develop the PPC disease than unbaited trees, while Erbilgin et al. [121] concluded that the transmission percentage and lesion length were positively correlated with the propagule load. Erbilgin et al. [121] also studied *P. carmeli* and found that adults artificially inoculated and confined to *P. radiata* branches transmitted the disease, therefore confirming their role as vectors under artificial conditions.

Species of the genus *Ips* (Coleoptera: Curculionidae) can cause large-scale damage to conifer forests with destructive outbreaks usually triggered by weakened trees due to fires, windstorms, drought or other pests or diseases [122]. Trees weakened by PPC infections can be vulnerable to attacks by *Ips* beetles [123], which subsequently act as vectors of *F. circinatum* [124,125]. In California, *Ips paraconfusus* Lanier, 1970, has been confirmed as a vector of PPC [4,5], and Erbilgin et al. [126] found *F. circinatum* on 10%–17% of the adults of *I. paraconfusus* and *Ips plastographus* (LeConte, 1868). Spores of the pathogen were found also inside the galleries and on the body of larval stages of *Ips mexicanus*

(Hopkins, 1905) and *I. paraconfusus* [5]. The contamination rates and propagule loads suggest a higher risk of PPC transmission taking place in spring and early summer [127].

Other bark beetles such as *Hylurgops* spp. and *Dendroctonus valens* LeConte, 1860 (Coleoptera: Curculionidae) are considered carriers of the PPC pathogen in California [128].

In Chile, *F. circinatum* has occasionally been isolated from nursery plants damaged by *Hylastes ater* (Paykull, 1800) [129], suggesting this insect could act as a carrier, with *Hylastes* spp. also considered as carriers in California [128].

The pathogen *F. circinatum* has been isolated from *Pissodes radiatae* (Hopkins, 1911) (Coleoptera: Curculionidae) collected with sticky and funnel traps near infected trees in California [30,128].

Similar results were reported by Storer et al. [127], who found a low phoresy rate of the PPC fungus on these weevils. Several other cone insects seem to facilitate the entrance of *F. circinatum* into the host plant as wounding agents. An association between the weevil *Pissodes nemorensis* (Germar, 1824) (Coleoptera: Curculionidae) and PPC has been suggested in the USA [114,130]. In South Africa, such association was predicted [131] based on a similar interaction previously demonstrated between *P. nemorensis* and *D. pinea* [132].

### 2.2.3. Cone Insects

*Fusarium circinatum* is known to infect cones [133,134], and therefore cone insects are potential vectors of PPC. Although there is no confirmation of cone insects vectoring *F. circinatum*, a few species were identified as carriers, namely *Conophthorus radiatae* (Hopkins, 1915) (Coleoptera: Curculionidae) and *Ernobius punctulatus* (LeConte, 1859) (Coleoptera: Anobiidae) in California [120,128,135]. *E. punctulatus* requires an entry tunnel previously made by another insect (e.g., *C. radiatae*) and transmission of inoculum between both species was already suggested by Hoover et al. [136]. Thus, *E. punctulatus* would not increase the infestation rate if *C. radiatae* had already introduced the pathogen into the cone. However, if the latter does not transmit the pathogen, *E. punctulatus* may enter the cone (using the entry tunnel made by the weevil) and introduce the pathogen [137]. Furthermore, cone beetles could promote a more efficient infection because this would take place deeply into the cone and, therefore, moisture on the cone surface would not be a requirement [138].

Conelet abortion and seed deterioration caused by PPC has also been associated with feeding by *L. corculus*, *Tetrya bipunctata* (Herrich-Schäffer, 1839) (Hemiptera: Scutelleridae) and *Laspeyresia* spp. (Lepidoptera: Tortricidae) [115], although these associations should be further clarified.

### 2.2.4. Sucking Insects

An association between spittlebugs and the PPC pathogen has been reported in California. In field samplings, Storer et al. [139] found that shoots with spittlebugs (*Aphrophora canadensis* Walley 1928) (Hemiptera: Aphrophoridae) feeding were more likely to develop PPC than shoots without feeding, while in a controlled experiment the incidence of PPC on shoots of potted *P. radiata* trees depended on the presence of spittlebugs [139]. Correll et al. [128] found the PPC pathogen on specimens of *Aphrophora permutata* Uhler 1875, suggesting these insects could act as carriers of the pathogen and not solely as wounding agents.

### 2.2.5. Predators

Some predatory insects are thought to be carriers of the PPC pathogen in California [4]. The genus *Lasconotus* (Coleoptera: Zopheridae) includes cylindrical beetles which predate on bark beetles in their galleries [140], where they may get the propagules of *F. circinatum* [141]. In a study performed by McNee et al. [135], the phoresy rate for *Lasconotus pertenuis* (Casey, 1890) in two locations in California (USA) was 37.5% ± 7.8% and 16.1% ± 1.9%. Storer et al. [127] mentioned that these beetles have the capacity to transmit the pathogen into bark beetle galleries free of the pathogen, although experiments fulfilling Leach's postulates [142] have not been conducted yet.

Larvae of *Medetera* spp. (Diptera: Dolichopodidae) are frequently found under the bark of trees, acting as bark beetle predators. However, only one reference has been found involving species from this genus in the spread of the PPC pathogen [127], and, once again, Leach's postulates have not been met yet.

*Enoclerus sphegeus* Fabricius 1787 (Coleoptera: Cleridae) is a predator of bark beetles which has been related to the spread of *F. circinatum* in California [127]. The authors report the potential of this species to transmit the pathogen into the bark beetle galleries free of the pathogen, similarly to *L. pertenuis* and *Medetera* spp. discussed above.

## 3. Insect-Mediated Spreading of PPC in Nurseries and Forests

The potential role of insects in spreading or facilitating the PPC differs according to the pest's capacity to damage pine seedlings and/or young and adult trees, host resistance, environment, and human activities [128]. In this section, we discuss the insect-mediated spreading of PPC in relation to the different phases of forest stand development, from seeds and seedlings to mature stands.

### 3.1. Nurseries

Nurseries play an essential role in the PPC spread at a regional scale, because the disease may be effectively dispersed with infected planting stock. In nurseries, potential sources of fungal inoculum are the soil and other growth media, used containers, weeds and irrigation water, as well as seeds.

Pine seeds are known to be prone to contamination by *F. circinatum* and transfer of seeds is a major pathway for the long-distance spreading of PPC [143]. The seeds may be infected early during their development: the fungus has been found to cause mortality of female flowers and mature strobili, and may infect and destroy gametophyte tissues of pine seeds [144]. Nevertheless, the same authors conclude that in many seed lots, the seed contamination by *F. circinatum* is restricted to the outer seed coat of otherwise healthy, viable seeds, and that actual infection and destruction of endosperm and embryos may occur only rarely. Cone and seed-associated insects can have strong impacts on seed production, but, in general, their influence in PPC dispersal is not well studied or is assumed to be secondary, as already discussed in the previous chapters.

Seeds that are contaminated in the forest or seed orchards may be brought to the nurseries to obtain seedlings. The nursery seedlings are particularly vulnerable for insects that damage the developing root system, with the succulent foliage also providing an attractive feeding substrate for defoliators. The development of seeds to seedlings can last one or two years, with the nursery plants exposed to a variety of stressful factors such as adverse soil proprieties, high air humidity promoting the growth of pathogenic fungi or insect pests. Sandy and silty soils with high humidity can be favourable for insects such as *Melolontha* spp. (Coleoptera: Scarabaeidae), the larvae of which forage underground on roots of different tree species. Other common scarabeid insects may be involved in spreading of PPC in nurseries, including species of the genera *Rhizotrogus, Amphimallon*, *Anomala*, *Polyphylla,* and *Phyllopertha*, which have polyphagous root eater larvae (grubs) and defoliator (on broadleaved species) adults. The root and collar eaters can easily get in contact with mycelium and spores of *Fusarium* on infected soil [145]. Root eaters can be abundant, and cause significant economic impacts in agriculture, horticulture, viticulture, and forestry, namely the cockchafers *Melolontha melolontha* (Linnaeus, 1758) and *Melolontha hippocastani* (Fabricius, 1801) [146]. In Poland, damage by grubs in nurseries and young plantations resulted in economic losses up to 7 M € in 2006 [147–149]. Other insects that cause damage in nurseries include wireworms (Coleoptera: Elateridae; e.g., genera *Agriotes* and *Selatosomus*), tenebrionids and phytophagous dipterans [146]. Wireworms of click beetles (Coleoptera: Elateridae), with the most common species belonging to the genus *Agriotes* in Europe, are less aggressive than grubs, yet important in nurseries where they damage the seedlings in early developmental stages [150]. Feeding damage by larvae of *Otiorhynchus sulcatus* (Fabricius, 1775) (Coleoptera: Curculionidae), which results in visible wilting and mortality, is frequently observed in nursery seedlings, while roots are also damaged by *Gryllotalpa gryllotalpa* (Linnaeus, 1758) (Orthoptera:

Gryllotalpidae)—one of the most conspicuous orthopteran forest pests [151]. Altogether, we can conclude that there is a widespread, abundant and diversified array of insect species, which can, directly or indirectly, favor the establishment and spreading of *F. circinatum* in European nurseries.

*3.2. Forests*

In many countries, the weevil *H. abietis* is the most destructive pest of pine plantations, as it can destroy up to 50%–100% of seedlings in recent plantation [152–156]. Weevils of the genus *Pissodes* are generally considered less aggressive, although *P. castaneus* can cause damage while feeding on the collar of young pine trees, whereas *Pissodes piniphilus* (Herbst, 1797) and *Pissodes pini* (Linnaeus, 1758) infest the aboveground parts of older trees [80]. Other common insects with high potential to spread PPC are scarabeid beetles of the genus *Melolontha*, which can devastate entire young plantations of coniferous species [157]. The regeneration and maturation feeding of *Hylastes* species (*H. ater* and *Hylastes cunicularius* (Erichson, 1836)) occurs around the root collar, killing seedlings, and sometimes destroying whole plantations [86,158]. Moreover, *Hylobius* spp. and *Hylastes* spp. may spread the disease at long distance, as these insects have been found at distances between 43 and 171 km of the nearest plantations [159,160].

Insects that infest weakened trees through regeneration feeding on twigs and buds usually do not cause major damage, but their larvae may severely damage the water and nutrient transporting tissues when feeding under the bark of the trees [161]. The importance of bark beetles and wood-boring insects increases on older forests, and while most species attack weakened trees as secondary pests, some are able to primarily attack pines. Climate stress (intense winds, severe droughts, fires, floods, etc.), other pests (including invasive species), and anthropogenic activities (unsustainable silviculture, pollution, etc.) are the most important factors which contribute to the occurrence of bark beetle outbreaks [162–165]. Shoot-feeding beetles infest the crown causing growth losses, and engraver beetles, with their associated fungi, can cause significant tree mortality [166].

Due to extreme weather conditions, bark beetle outbreaks have reached an unprecedented distribution and intensity in many countries, leading to severe alterations in forest ecosystem processes [5,123,167–171]. Pines have been affected by these outbreaks especially in areas where they have been planted out of their natural ecological (historical) range [172]. Recent dramatic outbreaks of *Dendroctonus ponderosae* (Hopkins, 1902) affected more than 1.2 M ha of *Pinus contorta* var. *latifolia* Engelmann in Central USA [173], whereas *Dendroctonus armandi* (Tsai and Li, 1959) has killed 300 M m$^3$ of *Pinus armandii* Franch. in China since the 1970s [174]. In Europe, severe attacks of *T. piniperda* has been reported to reduce forest productivity up to 45% of annual increment in Scandinavia [175], with losses of up to 72 M €/year in the Basque Country [176]. At the interface of the Mediterranean and Maritime zones, *I. sexdentatus* affected 37.9 M m$^3$ of *P. pinaster* forest after the passage of the hurricane "Klaus", with additional damages reported on *P. radiata* forests in the Basque Country [177]. In Russia, a large outbreak of the defoliator *Dendrolimus sibiricus* (Tschetverikov, 1908) (Lepidoptera: Lasiocampidae) affected 2.5 M ha of *Pinus sibirica* (Du Tour) and other conifers [165]. Forests affected by repeated attacks by pests and diseases, or from other abiotic-induced problems, are therefore more susceptible for *F. circinatum* establishment, not only by the lower vigor of the trees but also by the presence of multiple insects of various trophic guilds, which can, isolated or combined, act as vectors (*sensu lato*) of the PPC pathogen.

## 4. Spatial and Temporal Aspects of Insect-Mediated Transmission of PPC

A thorough consideration of the spatial and temporal aspects related to insect–host interactions may help to foresee the risk of insect-mediated spreading of PPC, and to design appropriate intervention measures and sustainable management strategies for nurseries, plantations and natural forests. In all these environments, the transmission of *F. circinatum* by insects occurs across different spatial and temporal barriers, which can be classified as "filters" *sensu* [178]. "Encounter filter" determines if,

to what extent, the pest or pathogen finds the potential host, and "compatibility filter" determines their success after the contact with a host is made.

The degree of spatial mobility of the carrier or vector insects is one of the crucial factors affecting the risk of insect-mediated spreading of *F. circinatum* to potential hosts at a landscape level. Pine shoot beetles (*Tomicus* spp.) can carry the spores of *F. circinatum* [179], which develop fast inside the breeding tunnels and infect larvae, pupae and adults [5]. The ability to feed on different shoots during their maturation-feeding phase favors the further spread of the fungus. They are also known to be good flyers, with an average dispersal distance ranging from 250 to 400 m, with some individuals being able to fly up to 2 km [180,181]. The mean flight distance for *Ips* spp. has been reported to be around 1.5–2 km, and a small percentage of adults can fly more than 5 km [182,183], while for *O. erosus* mark–recapture trials have suggested flight distances between 100 and 500 m, with some individuals recaptured up to 4–10 km from the release point [184]. The relatively large beetles of the genus *Monochamus* are also capable of moving long distances. Hernández et al. [185]; Más et al. [186]; David et al. [187] and Etxebeste et al. [188] found high within-population variability of flight performance in *M. galloprovincialis* with a mean distance of 16 km flown over the lifetime of the beetle. Maximum flight distance of about 5.5 km was reported for *Monochamus sutor* (Linnaeus, 1758), with mean distances ranging between 1.2 and 1.7 km [189].

Insect-mediated encounters between pines and *F. circinatum* are also strongly determined by the temporal and spatial patterns in the development of insect populations. Environmental factors, especially temperature, are key determinants of this synchrony [190]. Because insects are ectothermic, their development and reproduction are readily affected by changes in temperature regimes. For instance, voltinism may increase due to the climate-change related temperature elevations, potentially multiplying the encounters between pines and *F. circinatum* propagules. Multivoltine populations of *Ips typographus* (Linnaeus, 1758) have already been observed in spruce forests of high altitude and latitude areas [191]. The climate impact on voltinism is also illustrated by the case of *T. destruens*, which in the Mediterranean coastal pine forests has two breeding periods, one before winter and another from February to June [15,192].

Temporal patterns in host susceptibility are often related to phenological (seasonal) changes [193], e.g., bud burst [194] with the transition from juvenile phase to maturity. These patterns are often accompanied by variation in the nutritious quality and defensive chemistry and anatomy, which are a part of the "compatibility filter". For instance, Erbilgin and Colgan [195] studied *Pinus banksiana* (Lambert) responses to multiple enemies and defense inducers, reporting that although the constitutive monoterpenes were higher in the phloem of juvenile trees, the phloem of mature trees had a much higher magnitude of the defense induction. Changes in nutritional and defensive chemistry that occur in the host during the phenological development are expected to affect phytophagous insects, in particular species feeding or developing on annual shoots, such as shoot feeders and sucking insects. For instance, *R. buoliana*, which is particularly damaging on *P. nigra* and *P. sylvestris* [196], could be responsive to host phenological stage: the larvae, after an initial period spent mining in the needles, penetrate into the shoots through the buds and begin to bore a feeding gallery to complete their development [93]. To date, the only report on the association between *Rhyacionia* spp. and *F. circinatum* is by Matthews [116], who referred to seedlings in nurseries. Considering the feeding habits of the moth, its involvement in the spread of PPC could be mainly due to its potential role as a wounding agent.

Forest management can modify various spatiotemporal aspects related to insect-mediated spread of PPC. This could involve bottom-up effects on plant quality (e.g., through fertilization), influencing the "compatibility filter", or top-down impacts of stand characteristics (e.g., through thinning operations), influencing the habitats for the natural enemies of the carrier or vector insects [197]. Management for even-aged monocultures may provide insects with concentrated resources and thus effectively support local population growth, whereas the complexity of a mixed forest can make it more difficult for the insect pests to find the host trees [198]: thus, the species composition can effectively regulate

the "encounter filter". In general, by acknowledging the complex spatiotemporal aspects, e.g., through modelling approaches, we may be able to significantly improve our predictive and responsive capacities and develop targeted management strategies to suppress the insect-mediated spread of PPC.

## 5. Conclusions

- Numerous insect species that commonly occur in pine nurseries and forests throughout Europe and elsewhere have the potential to spread PPC as either vectors, carriers or wounding agents. Most of the evidence is, however, circumstantial and ambiguous, and in order to accurately model the spreading risk of PPC, more detailed information is required on the effective capacity of the different species to act as vectors. Such information can only be achieved through extensive field and laboratory studies.

- The importance of insect-mediated distribution of PPC also varies in different stages of forest development and forestry systems. The young seedlings, found in high densities in nurseries and plantations, attract different feeding guilds of insects than the old trees with large stem biomass and thick bark. Thorough understanding of the insect ecology and biology while they develop on trees in nurseries and in different stages of forest development will help to develop integrated pest management strategies to minimize the insect-mediated spreading of PPC. Effective pest control is important especially in nurseries, which have a crucial role in the regional spreading of PPC.

- It is clear that different feeding habits and life style of the insect species influence the risk of insect-mediated spreading of PPC. The phoretic rate, the ability to move between habitats (e.g., flight distances), seasonal timing, duration and location of feeding activities, fungal guilds involved in phoretic interactions, population fluctuations and type of damage/wounds caused by the insects differ markedly between species and feeding guilds, and should be carefully considered when modelling the risk of PPC spreading to new areas.

**Author Contributions:** Writing: M.F.-F., P.N., J.W., D.L.M., A.V.S., M.P., D.C., P.M.-Á., J.M.-G., E.J.M.-A., A.A., G.E.M.C., S.D.S. and C.Z.; review: M.F.-F., P.N., J.W. and J.J.D.; editing: M.F.-F. and A.A.

**Funding:** This study was made possible through the project "Etiology, Epidemiology and Control of *Fusarium circinatum*" sponsored by the Ministry of Rural, Marine, and Natural Environment, and with the support of the Government of Cantabria. This article is based upon work from COST Action FP1406 PINESTRENGTH (Pine pitch canker strategies for management of *Gibberella circinata* in greenhouses and forests) supported by COST (European Cooperation in Science and Technology) and project AGL2015-69370-R funded by MINECO and FEDER. This work is a contribution of URGENTpine (PTDC/AGR-FOR/2768/2014) funded by Fundação para a Ciência e a Tecnologia, I.P., through national funds, and the co-funding by the FEDER (POCI-01-0145-FEDER-016785), within the PT2020 Partnership Agreement and Compete 2020. Thanks for financial support are due to CESAM (UID/AMB/50017/2019), to FCT/MEC through national funds, and the co-funding by the FEDER, within the PT2020 Partnership Agreement and Compete 2020. FCT also supported J.M.-G. (SFRH/BPD/122928/2016). P.N. would like to thank funding by the Fundação para a Ciência e Tecnologia—FCT (contract IF/00471/2013/CP1203/CT0001), Portugal. D.L.M.'s contribution was also supported by The Russian Foundation for Basic Research (Grant No. 17-04-01486). A.V.S.'s contribution was also supported by the Center for collective use of scientific equipment "Renewable resources, energy sources, new materials and biotechnology (SPbFTU)" (Project 2019-0420).

**Acknowledgments:** We want to thank Martin Muller, Jaime Aguayo, Katarina Adamcikova, Kaljo Voolma, Stanley Freeman, Panos Petrakis, and Svetlana Markovskaja for their contribution in the collection of references related to the PPC disease.

**Conflicts of Interest:** The authors declare no conflict of interest.

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
