# Peer review of "Pine Pitch Canker and Insects: Relationships and Implications for Disease Spread in Europe"

_forests, doi:10.3390/f10080627_

Round 1

Reviewer 1 Report

The authors present a review article summarizing the current literature available on insects associated with pines and their known and/or possible roles as disseminators of pine pitch canker (PPC) disease. The review is divided in five main sections: 1. A general introduction, 2. Interactions ofF. circinatum with insects in European and non-European countries, 3. Insect-mediated spreading of PPC in nurseries and forests, 4. Spatial and temporal aspects of insect-mediated transmission of PPC, and 5. Conclusions.

The article is quite thorough and represents a significant contribution to the current knowledge of the role of insects involved in the dissemination of pine pitch canker disease. The review mainly focuses on the second and third sections, further dividing these sections into types of insects based on their role as feeders. In general, the article is well written and has good flow.  I therefore consider that this article is suitable for publication after some minor revisions. I am including some detailed comments by section in the attached document.

Author Response

Dear reviewer,

Thank you very much for your comments, sugestions and so on that help us to improve the manuscript. As you can see, all your suggestions have been included in the text and we would like also to answer to your questions (see below).

Best regards,

Mercedes Fernández

---------------------------------------------------------------------------------------------------

Reviewer’s comments:

The authors present a review article summarizing the current literature available on insects associated with pines and their known and/or possible roles as disseminators of pine pitch canker (PPC) disease. The review is divided in five main sections: 1. A general introduction, 2. Interactions of F. circinatum with insects in European and non-European countries, 3. Insect-mediated spreading of PPC in nurseries and forests, 4. Spatial and temporal aspects of insect-mediated transmission of PPC, and 5. Conclusions.

The article is quite thorough and represents a significant contribution to the current knowledge of the role of insects involved in the dissemination of pine pitch canker disease. The review mainly focuses on the second and third sections, further dividing these sections into types of insects based on their role as feeders. In general, the article is well written and has good flow.  I therefore consider that this article is suitable for publication after some minor revisions. I am including some detailed comments by section below:

Title: Although using “the spreading” is linguistically correct as a noun, it sounds a bit awkward. How about substituting “the spreading” for “dissemination”?

-        We agree and, in order to avoid missunderstandings we change the title by:

Pine pitch canker and insects: Relationships and implications for the spreading of the disease.

Abstract: Summarizes adequately the paper. Sentence on lines 42-44 is not clear. It needs to be reworded to better convey the ideas that are presented.

-        We agree. Changed.

Introduction: Provides a general background on pitch canker disease and the insects associated with it and with pine plantations. However, in some instances, references are needed to back the statements presented.

-        References were added in the manuscript.

See attached pictures for some comments:

Section 2. Interactions of Fusarium circinatum with insects: This section is a comprehensive description of different species of insects associated to F. circinatum within Europe (Section 2.1) and in non-European countries (Section 2.2). In some instances, this section also needs some citations to back some of the statements. Also, in Section 2.2., most examples presented as non-European countries mainly describe situations in North America, such as California and Southeast United States. While this is expected, as pitch canker disease is believed to have originated from Central and North America, and most studies had been conducted in the Southeast United States since the mid-20th century, more recent cases of pitch canker disease have been described in other continents, such as Asia ( Kobayashi et al. 1989), Oceania (Dick, 1998), and Africa (Coutinho et al., 2007). Are there any more recent studies of this disease in these areas that have looked at insects related to pitch canker disease? I think that this would be key to this review, if the authors wish to provide an exhaustive, global view of putative insect-mediated transmission of this disease.  Otherwise, the authors should indicate as they introduce this subsection, why it is focused mainly in North America.

-        We appreciate the comment. The reviewer is correct, pine pitch canker (PPC) is now present in many other countries beyond USA (e.g. South Africa, Japon, South Korea, Mexico, Chile, Uruguay, Colombia, Brazil, etc). However, to our knowledge, the available information regarding insects interacting with the pathogen outside of Europe is limited to studies carried out in USA, and, to a lesser extent, from Chile and South Africa. On the other hand, although PPC has not been reported in Oceania, Dick (1998) and Brockerhoff et al. (2016) reported some potential vectors of PPC present in New Zealand. But taking into account that this information was already collated by these authors and no new relevant information is available, we have preferred to focus only on the interactions F. circinatum/insects in those countries where PPC is already presented.

-        We have add a paragraph in the manuscript in order to try to clarify this aspect.

The rest of the paper only requires minor edits. I am attaching some of my edits in the pictures below. Rather than going line-by-line, I think it’s most efficient to track as pictures, as long as my handwriting and picture quality are not an impediment.

-          No problem. I changed all of them. We also added  the references you asked for.

Please ignore my personal notes in the bottom of page 15.

-        O.K. Thank you very much for you suggestions in order to improve the manuscript.

Reviewer 2 Report

The aim of this manuscript is to review the taxonomic and ecological diversity of insect species with high potential association with Fusarium circinatum in Europe and elsewhere. F. circinatum is an ascomycete causing pine pitch canker, which is one of the most destructive fungal diseases of pines worldwide. The fungus enters the plant normally by wounds open by insects or other biotic or abiotic agents, although infection without wound may also occur. The infection facilitated by insects is well established and from this point of view the review could be interesting.

However, I cannot recommend this paper for publication. From my perspective, the current manuscript offer no novel insights; the topic addressed have already been discussed extensively elsewhere (Storer et al. 1998, Brockerhoff  et al. 2016, Flores Pacheco 2017, Selikhovkin et al. 2018, among others). Even the latest author is coauthor of this review and his previous work is no cited here.

The strength of the paper could be a revision focusing in the European species, however the few species positively related with F. circinatum in Europe have already been published and reviewed (Romon et al. 2007, Bezos et al. 2009, Bezos et al. 2017, Flores-Pacheco 2017). The rest of insects listed here are mostly assumptions of the authors, based in some biological characteristic of the species, which makes the paper highly speculative.

The authors make that list because they propose a definition of vectors sensu lato with which I have problems in agreeing. I may accept within vectors sensu lato “carriers” because they carry inoculum from diseased trees, although the ability to transfer it to healthy trees is not proved yet.  Nevertheless, they also include in the definition insects that made wounds in trees and therefore may facilitate the penetration of the fungi within the living tissue. However, I do not think that these insects can be considered vectors. A vector has to have an association with the diseased plant, carrying the pathogen, visiting susceptible plant hosts, and must be capable of successfully transmitting the pathogen to healthy plants.

Just open a wound is not enough to be vector a disease, even if wounding is critical for infection. Sakamoto and Gordon (2006) show that not all types of wounds result in pitch canker. And Ganley (2007), suggest that for insects, infections may only occur from 1) the creation of wounds deep enough to provide adequate moisture, 2) wounds created by insects that also vector the pathogen (i.e. spores present at wounds before wound can dry out), or 3) wounds that do not result in substantial resin production.

Many more factors must be taking into account when considering fungus-insect-plant interactions. For instance Lombardero et al. (2019) suggest that the defensive reaction of the pine tree against Tomicus spp. inhibits the growth of the fungi (Line 154). Therefore even a species considered a plausible vector may have a negative influence in the epidemiology of the disease. What we know about all the other insects cited here?  And why the authors list these particular insects and not others with similar biology? Lieutier et al. (2004) provided a longer list of bark and wood boring insects than the ones listed here.

Why add scale insects living in the needles? If they can facilitate the entrance of the fungus through the needles, why not all the defoliators?  They also open wounds in needles.

The paper continues with a section reviewing insects from non-European countries, which I think is quite limited because they mainly review what happens in the US and not in other geographic areas where the fungus is established.

The next sections, 3 and 4, do not add anything more to the paper. In my opinion, the review does not provide a framework of the multiple factors that regulate the insect-host interactions and determine the success of the infection, as the authors state in the abstract.

Other  comments:

Line 121. The text reads: Other insect species can be considered potential vectors (sensu lato) of F. circinatum in Europe, and are summarized in Table 1. However Table 1 included also the species cited in the previous paragraph too. So either rephrase the sentence or delete the species already cited from the table.

Table 1:

-  What are the criteria to consider a species “carrier” or “probably carrier”?  Many of the species considered carriers were not previously related with F. circinatum .

- Tomicus spp is considered a vector in the table although his role vectoring the disease is no totally proved yet. But if the authors are just guessing for most of the insects, why Monochamus is considered only a wounding agent if it has a biology similar to Tomicus spp?; lays eggs in weakened trees and feed in healthy shoots.

Line 156: I think authors should clarify why Tomicus should be considered important vectors of Fusarium, especially if previously they introduce this species as a plausible vector with low phoresy rates and continue citing at least 3 papers showing that F. circinatum was not isolated from Tomicus in stands affected with diseases.

Line 176: D. sylvestrella is mainly a stem borer (common name: Pine Stem Borer)

Line 193: Engraver beetles seems to me a different classification than the rest of categories (shoot feeders, twig beetle, bark and root borers or cone beetles). "Engraver beetles" does not refer to the place where they feed but to the type of galleries that they make. In my understanding is a common name for Ips and related species, so may be is more reasonable to refer this subheading as "bark beetles".

Line 221: Do you mean “introduced” or “is distributed”?

Line 229: Dendroctonus micans may attack pines but usually prefers spruce (Common name: Great spruce bark beetle)

Line 230: Why no other borers as Rhagium, Arhopalus, Spondylis, Sirex, Melanophila,  etc?

Line 239: Check the reference for North America citation.

-          May be this and the previous comment of D. valens fit better in the non-European countries section.

Line 263: Why not other cone borers as Dioryctria mendacella?

Line 290: Why not other suckers as spittlebugs (Haematoloma dorsatum)?

Line 313: What do you mean with “In Europe shoot feeders are not consider wounding agents”?

Line 440: There is some reference of these insects attacking plants affected by Fusarium in the field?.

Line 461: I do not see the intention of the authors in the paragraph of extreme weather condition. Does Fusarium circinatum colonize preferentially weakened trees?.

Author Response

Dear reviewer,

I would like to thank you very much for your comments, suggestions and so on that are really helpfull in order to improve the manuscript. We took all of them into account and we would like to answer to you. Please, see below.

Kind regards,

Mercedes Fernández

------------------------------------------------------------------------------------------------------------

The aim of this manuscript is to review the taxonomic and ecological diversity of insect species with high potential association with Fusarium circinatum in Europe and elsewhere. F. circinatum is an ascomycete causing pine pitch canker, which is one of the most destructive fungal diseases of pines worldwide. The fungus enters the plant normally by wounds open by insects or other biotic or abiotic agents, although infection without wound may also occur. The infection facilitated by insects is well established and from this point of view the review could be interesting.

However, I cannot recommend this paper for publication. From my perspective, the current manuscript offer no novel insights; the topic addressed have already been discussed extensively elsewhere (Storer et al. 1998, Brockerhoff  et al. 2016, Flores Pacheco 2017, Selikhovkin et al. 2018, among others). Even the latest author is coauthor of this review and his previous work is no cited here.

The strength of the paper could be a revision focusing in the European species, however the few species positively related with F. circinatum in Europe have already been published and reviewed (Romon et al. 2007, Bezos et al. 2009, Bezos et al. 2017, Flores-Pacheco 2017). The rest of insects listed here are mostly assumptions of the authors, based in some biological characteristic of the species, which makes the paper highly speculative.

·       Storer et al, 1998 is quite old. The others are newer and we could check and refer to, and specify what is different in this review.

·       Selihkovkin et al., 2018: this article is narrower focus on Russia; and no expansion to management and control options are presented. The reason why it is not included is only that this paper was published after having written this first version of the paper but, anyway, now, it has been included.  

·       Flores Pacheco, 2017 is a good review but it was published in a review of Nicaragua and only in Spanish.

-        Anyway, we think that this review is a little bit different to the previous mentioned because it has a pan European focus, it has a broader range as well as a more systematic presentation of this. We would also to clarify that we are presenting potential aspects of the insects in the transmission of the disease. We also like to focus in those insects groups where a more urgent research is needed to better understand the transmission way of the pathogen. One of our goals, that we would like to specify, is that future research are needed.

The authors make that list because they propose a definition of vectors sensu lato with which I have problems in agreeing. I may accept within vectors sensu lato “carriers” because they carry inoculum from diseased trees, although the ability to transfer it to healthy trees is not proved yet.  Nevertheless, they also include in the definition insects that made wounds in trees and therefore may facilitate the penetration of the fungi within the living tissue. However, I do not think that these insects can be considered vectors. A vector has to have an association with the diseased plant, carrying the pathogen, visiting susceptible plant hosts, and must be capable of successfully transmitting the pathogen to healthy plants.

-        We would like to clarify that the use of “vectors sensu lato” is only a general way to simplify and to organize the three categories of insects. In order to avoid misunderstanding, we eliminated it.  

Just open a wound is not enough to be vector a disease, even if wounding is critical for infection. Sakamoto and Gordon (2006) show that not all types of wounds result in pitch canker. And Ganley (2007), suggest that for insects, infections may only occur from 1) the creation of wounds deep enough to provide adequate moisture, 2) wounds created by insects that also vector the pathogen (i.e. spores present at wounds before wound can dry out), or 3) wounds that do not result in substantial resin production.

-        We agree but when there is no other evidences (no scientific demonstrations of insects carrying spores) we can suppose that all these insects could act at least, as wounding agents. Anyway, we add those references and a little explanation to the text in order to clarify.

Many more factors must be taking into account when considering fungus-insect-plant interactions. For instance Lombardero et al. (2019) suggest that the defensive reaction of the pine tree against Tomicus spp. inhibits the growth of the fungi (Line 154). Therefore even a species considered a plausible vector may have a negative influence in the epidemiology of the disease. What we know about all the other insects cited here? 

-        Thank you very much for this suggestion. However, this relationship is even harder to demonstrate than a relation between an insect and F. circinatum spores. Therefore, we will not deal with this option more in this particular review. Also, we can mention possibilities of different trophic cascades as potential factors affecting the outcome but we don’t have possibility to cover everything (we have to limit the scope…).

And why the authors list these particular insects and not others with similar biology? Lieutier et al. (2004) provided a longer list of bark and wood boring insects than the ones listed here.

-        As said, we were not able to include everything, but hope to provide examples from different scenarios…and we chose those one we think there are more representatives.

Why add scale insects living in the needles? If they can facilitate the entrance of the fungus through the needles, why not all the defoliators?  They also open wounds in needles.

-        We changed some of the insects´categories, e.g. shoot and foliage feeders to include defoliators such as Rhyacionia spp.

The paper continues with a section reviewing insects from non-European countries, which I think is quite limited because they mainly review what happens in the US and not in other geographic areas where the fungus is established.

-        We agree with this opinion and we will add an explanation that the examples are mainly from US, because US is a good reference point taking into account the relationship between the disease and the insects. American papers related with this, were the first ones published and were the basis for the following studies carried out in other countries. Only some few information from Chile and South Africa were found focused on insects.

The next sections, 3 and 4, do not add anything more to the paper. In my opinion, the review does not provide a framework of the multiple factors that regulate the insect-host interactions and determine the success of the infection, as the authors state in the abstract.

-        We would like to emphasize that we consider it important for development of proactive control and management strategies to analyse, using a holistic approach, the potential of different insect species to disseminate the spores and thus contribute to the risk of PPC spread in different conditions. Also, if we can emphasize more clearly a few urgent knowledge gaps raising from our analysis, the contribution to future forest health would be strong.

Other  comments:

Line 121. The text reads: Other insect species can be considered potential vectors (sensu lato) of F. circinatum in Europe, and are summarized in Table 1. However Table 1 included also the species cited in the previous paragraph too. So either rephrase the sentence or delete the species already cited from the table.

-      We are agree. It was eliminated and the table was reorganized.

Table 1:  What are the criteria to consider a species “carrier” or “probably carrier”?  Many of the species considered carriers were not previously related with F. circinatum.

- When some authors have found spores of F. circinatum on the insect, we consider it as a carrier. However, when nobody has noticed the spores nor more info is available but, the species is very close (ecobiology) to another one that has been demonstrated to be a carrier, we says as “probably carrier”. Anyway, We will eliminate “probably” from the table.

- Tomicus spp is considered a vector in the table although his role vectoring the disease is no totally proved yet. But if the authors are just guessing for most of the insects, why Monochamus is considered only a wounding agent if it has a biology similar to Tomicus spp?; lays eggs in weakened trees and feed in healthy shoots.

- We are agree that the feeding habits of Monochamus is the same as Tomicus but the point is that nobody has found spores of F. circinatum on its body still now and, we cannot consider it then, as a vector or a carrier, only as a wounding agent. 

Line 156: I think authors should clarify why Tomicus should be considered important vectors of Fusarium, especially if previously they introduce this species as a plausible vector with low phoresy rates and continue citing at least 3 papers showing that F. circinatum was not isolated from Tomicus in stands affected with diseases.

- We agree with you, you are true but, these species have a widespread distribution, they are important forest pests in several European countries and each individual, consume several healthy shoots for their maturation and regeneration feeding, causing a high number of wounds in healthy trees, even if the number of spores is low, if there is an epidemic situation and millions of beetles thaf feeding on 4-5 shoots/each, the possibility of transmission to healthy pines is higher. This is the reason why we consider them so important.

Line 176: D. sylvestrella is mainly a stem borer (common name: Pine Stem Borer).

-        We agree. We will eliminate it.

Line 193: Engraver beetles seems to me a different classification than the rest of categories (shoot feeders, twig beetle, bark and root borers or cone beetles). "Engraver beetles" does not refer to the place where they feed but to the type of galleries that they make. In my understanding is a common name for Ips and related species, so may be is more reasonable to refer this subheading as "bark beetles".

-        We agree. We will change it and classify them as bark beetles and wood borers.

Line 221: Do you mean “introduced” or “is distributed”?

-        Distributed. Changed.

Line 229: Dendroctonus micans may attack pines but usually prefers spruce (Common name: Great spruce bark beetle)

-        We agree. Both species will be eliminate from this part.

Line 230: Why no other borers as Rhagium, Arhopalus, Spondylis, Sirex, Melanophila,  etc?

-        We were not able to include everything, and we chose those species we consider more representatives or those ones that could have any close relationship with the disease. In the case of the wood borers, no information exists about this relationship, may be because this kind of niche is not so suitable for the spores of the fungi (not too much moisture in dead wood).

Line 239: Check the reference for North America citation.

- Changed.

-          May be this and the previous comment of D. valens fit better in the non-European countries section.

- We agree. Changed.

Line 263: Why not other cone borers as Dioryctria mendacella?

-        We were not able to include everything, and we chose those species we consider more representatives. Anyway, we included it in the text.

Line 290: Why not other suckers as spittlebugs (Haematoloma dorsatum)?

-        We were not able to include everything, but we included now it as another example of this insect category.

Line 313: What do you mean with “In Europe shoot feeders are not consider wounding agents”?

-        Changed.

Line 440: There is some reference of these insects attacking plants affected by Fusarium in the field?.

In this paragraph we revised the major insect pests of the pines and we highlighted the possibility of these insects to carry (vectorise) spores of Fusarium circinatum and infect (healthy) new seedlings / trees (and stands) not the possibility that these insects to affect the forest already infected with Fusarium. Also, there is no evidence (not yet) of these insects to attack plants already affected by Fusarium in the field.

Line 461: I do not see the intention of the authors in the paragraph of extreme weather condition. Does Fusarium circinatum colonize preferentially weakened trees?.

According to Furniss and Carolin (1977), Fox et al. (1990, 1991) and Storer et al. (2002a) Ips spp. are attracted to stressed trees thus, they are more likely to be involved in killing already weakened trees and spreading the disease (PPC) to adjacent trees, rather than initiating infections in uninfected, healthy stands.

The new references will be added to the document.

Reviewer 3 Report

Comments:

The paper is an excellent synthesis of the current state of knowledge about the interactions established between the fungus Fusarium circinatum and its insect sensu lato vectors written comprehensibly. The work presented stimulates the curiosity and point out the necessity of studied the hypotheses suggested.

Questions:

Is more correct   use the name Diplodia pinea or is synonymous Sphaeropsis sapinea?

Suggestions:

For an easy read, I suggest maintaining the order in which vectors sensu lato are characterized on lines 67-68 (wounding agents; carriers and vectors) in lines 89-97 instead the order presented (vectors, carriers and wounding agents).

Author Response

Dear referee,

Thank you very much for your suggestions.

I would like to answer your question: Is more correct use the name Diplodia pinea or is synonymous Sphaeropsis sapinea?

They are synonyms and both used commonly I think, but I notice that EPPO seems to recommend Diplodia pinea https://gd.eppo.int/taxon/DIPDPI so I think we could refer to that. Also, Luchi et al (to which we refer to use that name.

Suggestions:

For an easy read, I suggest maintaining the order in which vectors sensu lato are characterized on lines 67-68 (wounding agents; carriers and vectors) in lines 89-97 instead the order presented (vectors, carriers and wounding agents).

We agree. We already changed it in the manuscript.

Kind regards,

Mercedes Fernández

Reviewer 4 Report

The paper entitled "Pitch canker disease and insects: Relationships and implications for the spreading" is a review manuscript aimed to provides a wideview of the multiple factors that regulate the insect-host interactions that playing in the success of Fusarium circitatum infection.

Authors revised 201 bibliographic sources, covering 20 insect taxa (species and genus) that could be associated with F. circinatum and their possible transmission pathways. This result is summarized a extensed table, constituiting the most relevant result of this revision, and could be an usefull tool for consult.

This table is accompained of a text that show a deeply bibliographyc revission that constituted the true corpus of this manuscript. No figures has been incuded in the work.

However, and as specialist in the Tomicus genus, I was found some major considerations on Shoot feeders part:

1.- Authors assert in the lines 127-128 that "genus Tomicusare the most important vectors of PPC in Spain", citing Bezós et al (2015) when this paper works exclusively on Tomicus piniperda. To extend this vector ability to the other species is an oppinion that should be confirmed by an specific research.

2.- Next, authors revise the distributions of the three European Tomicusspecies, according the chapitre of Lieutier et al (2015), on the Genus Tomicus, citing"...T. minorand T. piniperda are widespread in Northern and Central Europe, T. destruensis a thermophilous species found in the coastal areas of the circum-Mediterranean countries.". But this sentence is a simpification that could inducces misunderstading on true distribution of this species. Accordingly the same paper (Lieutier et al 2015) and others (i.e: Kerdelhué et al 2002,  Gallego et al 2004, Vasconcellos et al 2006, Sánchez-García et al 2014), T. destruens is widely distributed in Iberian Peninsula and Portugal, but not only in Mediterranean coasts, including Bay of Biscay coasts and the french territory of The Landes. In my oppinion this paragraph needs more precission.

3.- I am in totaly desagreement with the sentence of 135-137 lines "...outbreaks of T. destruenshave  been reported in pine forests affected by consecutive years of drought and heavy defoliation by Thaumetopea pytiocampa...", in accordance with Sousa et al (2011) and Naves et al (2017). This is an asseveration with strongs implications in forest dinamic and management. Surprisingly, first paper do not treat onThaumetopoea-Tomicus. I read it and only cite T. Pityocampa andT. destruens in a paragraph of introduction and without  relating these interactions. The second reference (Naves et al 2017) is a national congress contribution that only cites T. destruensand T. pytiocampa as damage agents, between others agents, in a portuguese locality, but do not relate any interaction between both species. As I say before, this asseveration is a mistaken thopic, never evidenced, that can not repeat in a scientiphic paper.

4.- In this way, in lines 144-146, a sencence afirm on Tomicusshoot feeding: "The recurrent damage of the apical shoots disturbs the physiological balance in the trees, causing deformations and  predisposing them to subsequent attacks by Tomicusor other related pests.", according Pennacchio et al 2005. As in the previous consideration, this have important implications inTomicus management, taken into acount that in 25 years working on this genus, no predisposition between shoot and bark phase could be evidenced. I also read this paper,  aimed to enumerate the xylophagous species ofPinus pinaster stands attacked by Matsucoccus feitaduyin two Italian regions, and only a sentence on this predisposition has been found, without any scientific demostration.

5.- In line 509, a third generation of T. destruensin Northern Africa is cited, in accordance with Ben Jaama et al (2007) and Graf et al (1994). As in before considerations, I read the Ben Jaama et al  paper and I do not found any reference to this third T. destruens generation, in a paper aimed to assess virulence of phytopathogenic fungi associated to T. destruensand O. erosus.The second paper is not avaliable and I could not read it, although references to a this generation has been not cited never, even in the review of Lieutier et al (2015).

Minor Considerations

Line 183: the use of "secondary forest insect" is not the more appropiate in my oppinion. Expression as "non-aggresive" is more usefull, because the implication in tree mortality are more clearly.

Line 496: Revision on fligh distances of M. galloprovincialisis too scarce, taken into account that specific papers as Torres-Vila et al (2014), Hernandez et al (2011), Etxeveste et al (2016), Mas et al (2013) are easely found in a bibliography search.

In conclusion, I think that the paper is suitable for publication, but I found important inaccuracies at least in the theme that I am specialist, the Tomicus genus. This inaccuracies are based  on false thopics of forest pest, commonly extended between non-specialiced people, and incorectly supported by a bibliography that, or do not cover this thopics or repeat this thopics, but without scientific demostration.

So I recommend a new revision of this bibibliography in order to avoid include non scientific asseverations.

Author Response

Dear reviewer,

Thank you very much for all your comments, suggestions and so on that help to improve the manuscript. See below, our answers to your comments.

Kind regards,

Mercedes Fernández

----------------------------------------------------------------------------

1.- Authors assert in the lines 127-128 that "genus Tomicus are the most important vectors of PPC in Spain", citing Bezós et al (2015) when this paper works exclusively on Tomicus piniperda. To extend this vector ability to the other species is an opinion that should be confirmed by an specific research.

- We agree. Changed in the text and we mentioned the info related to T. piniperda.

2.- Next, authors revise the distributions of the three European Tomicus species, according the chapitre of Lieutier et al (2015), on the Genus Tomicus, citing"...T. minor and T. piniperda are widespread in Northern and Central Europe, T. destruens is a thermophilous species found in the coastal areas of the circum-Mediterranean countries.". But this sentence is a simplification that could induces misunderstanding on true distribution of this species. Accordingly the same paper (Lieutier et al 2015) and others (i.e: Kerdelhué et al 2002,  Gallego et al 2004, Vasconcellos et al 2006, Sánchez-García et al 2014), T. destruens is widely distributed in Iberian Peninsula and Portugal, but not only in Mediterranean coasts, including Bay of Biscay coasts and the french territory of The Landes. In my opinion this paragraph needs more precision.

- We agree and we changed this paragraph as follow: “In Spain, T. piniperda is the most plausible vector of PPC and, together with T. destruens, these species are widely distributed in Iberia Peninsula and Portugal”.

- We wrote this because in Lieutier et al (2015), we can read: “Tomicus destruens is strictly distributed in the coastal areas of the circum-Mediterranean countries and Portugal”.

3.- I am in totaly desagreement with the sentence of 135-137 lines "...outbreaks of T. destruens have  been reported in pine forests affected by consecutive years of drought and heavy defoliation by Thaumetopea pytiocampa...", in accordance with Sousa et al (2011) and Naves et al (2017). This is an asseveration with strongs implications in forest dinamic and management. Surprisingly, first paper do not treat on Thaumetopoea-Tomicus. I read it and only cite T. Pityocampa and T. destruens in a paragraph of introduction and without  relating these interactions. The second reference (Naves et al 2017) is a national congress contribution that only cites T. destruens and T. pytiocampa as damage agents, between others agents, in a portuguese locality, but do not relate any interaction between both species. As I say before, this asseveration is a mistaken topic, never evidenced, that cannot repeat in a scientific paper.

- Taking your explanations into account, we eliminated these two references.

4.- In this way, in lines 144-146, a sencence afirm on Tomicus shoot feeding: "The recurrent damage of the apical shoots disturbs the physiological balance in the trees, causing deformations and  predisposing them to subsequent attacks by Tomicus or other related pests.", according Pennacchio et al 2005. As in the previous consideration, this have important implications in Tomicus management, taken into account that in 25 years working on this genus, no predisposition between shoot and bark phase could be evidenced. I also read this paper, aimed to enumerate the xylophagous species of Pinus pinaster stands attacked by Matsucoccus feitaduy in two Italian regions, and only a sentence on this predisposition has been found, without any scientific demonstration.

- Changed in the manuscript and eliminated the “predisposition” part.

5.- In line 509, a third generation of T. destruens in Northern Africa is cited, in accordance with Ben Jaama et al (2007) and Graf et al (1994). As in before considerations, I read the Ben Jaama et al  paper and I do not found any reference to this third T. destruens generation, in a paper aimed to assess virulence of phytopathogenic fungi associated to T. destruensand O. erosus.The second paper is not avaliable and I could not read it, although references to a this generation has been not cited never, even in the review of Lieutier et al (2015).

- I agree with you. It sounds quite strange and I also checked the reference and no information exits. It was eliminated from the text as well as these related references.

Minor Considerations                                          

Line 183: the use of "secondary forest insect" is not the more appropiate in my oppinion. Expression as "non-aggresive" is more usefull, because the implication in tree mortality are more clearly.

-        I agree. Changed.

Line 496: Revision on fligh distances of M. galloprovincialisis too scarce, taken into account that specific papers as Torres-Vila et al (2014), Hernandez et al (2011), Etxeveste et al (2016), Mas et al (2013) are easely found in a bibliography search.

-        I added more new references to the text.

In conclusion, I think that the paper is suitable for publication, but I found important inaccuracies at least in the theme that I am specialist, the Tomicus genus. This inaccuracies are based  on false thopics of forest pest, commonly extended between non-specialiced people, and incorectly supported by a bibliography that, or do not cover this thopics or repeat this thopics, but without scientific demostration.

So I recommend a new revision of this bibibliography in order to avoid include non scientific asseverations.